# Ultrafast isomerization in acetylene dication after carbon K-shell ionization

Zheng Li[1,2], Ludger Inhester[3,4], Chelsea Liekhus-Schmaltz[1,5], Basile F.E. Curchod [1,2], James W. Snyder Jr.[1,2], Nikita Medvedev[3,6,7], James Cryan[1], Timur Osipov[1], Stefan Pabst [8], Oriol Vendrell [9], Phil Bucksbaum[1,5] & Todd J. Martinez [1,2]

Ultrafast proton migration and isomerization are key processes for acetylene and its ions. However, the mechanism for ultrafast isomerization of acetylene $[HCCH]^{2+}$ to vinylidene $[H_2CC]^{2+}$ dication remains nebulous. Theoretical studies show a large potential barrier ( > 2 eV) for isomerization on low-lying dicationic states, implying picosecond or longer isomerization timescales. However, a recent experiment at a femtosecond X-ray free-electron laser suggests sub-100 fs isomerization. Here we address this contradiction with a complete theoretical study of the dynamics of acetylene dication produced by Auger decay after X-ray photoionization of the carbon atom K shell. We find no sub-100 fs isomerization, while reproducing the salient features of the time-resolved Coulomb imaging experiment. This work resolves the seeming contradiction between experiment and theory and also calls for careful interpretation of structural information from the widely applied Coulomb momentum imaging method.

[1] SLAC National Accelerator Laboratory, 2575 Sand Hill Road, Menlo Park, California 94025, USA. [2] Department of Chemistry and the PULSE Institute, Stanford University, 333 Campus Drive, Stanford, California 94305, USA. [3] Center for Free Electron Laser Science, Deutsches Elektronen-Synchrotron, Notkestraße 85, D-22607 Hamburg, Germany. [4] Hamburg Center for Ultrafast Imaging, Luruper Chaussee 149, D-22761 Hamburg, Germany. [5] Department of Physics, Stanford University, 382 Via Pueblo Mall, Stanford, California 94305, USA. [6] Department of Radiation and Chemical Physics, Institute of Physics, Czech Academy of Sciences, Na Slovance 2, 182 21 Prague 8, Czech Republic. [7] Laser Plasma Department, Institute of Plasma Physics, Czech Academy of Sciences, Za Slovankou 3, 182 00 Prague 8, Czech Republic. [8] Harvard-Smithsonian Center for Astrophysics, 60 Garden Street, Cambridge, Massachusetts 02138, USA. [9] Department of Physics and Astronomy, Aarhus University, Ny Munkegade 120, 8000 Aarhus, Denmark. Correspondence and requests for materials should be addressed to T.J.M. (email: toddjmartinez@gmail.com)

Acetylene ($C_2H_2$) in neutral and ionic forms is an important species in combustion and atmospheric chemistry, and in the interstellar medium. The vinylidene isomer is an important intermediate in many reactions involving $C_2H_2$[1–8]. However, unlike in the neutral and cationic species[6–11], the pathway for isomerization of the acetylene dication, consisting of hydrogen migration from $[HCCH]^{2+}$ to $[H_2CC]^{2+}$, remains largely unresolved. The reason for this disparity is an apparent contradiction between theory and experiment, prompting numerous studies[5, 12–18]. Arguments on both sides can be summed up as follows. The experimental synchrotron data concludes that isomerization occurs on the low lying dicationic states $^1\Sigma_g$ and $^1\Delta_g$ with vacancies $1\pi_u^{-2}$, while deprotonation and symmetric breakup occur on the higher lying $^1\Pi_u$ states with $1\pi_u^{-1}3\sigma_g^{-1}$ character[13, 19]. In addition, photoelectron-photoion momentum spectroscopy experiments[12, 13] suggest that ultrafast hydrogen migration occurs in <100 fs. A third separate XFEL experiment at linac coherent light source was interpreted to show the existence of significant hydrogen migration within 100 fs[5]. These three pieces of evidence would seem to indicate that isomerization proceeds on the low-lying dicationic states on a sub-100 fs timescale.

In contrast to this experimental evidence, ab initio electronic structure calculations predict an isomerization barrier of ~2 eV on the $1\pi_u^{-2}$ double hole states[14, 15]. Isomerization over such a large barrier would be highly unlikely to occur on the femtosecond timescale and is expected to be orders of magnitude slower than the observed 100 fs isomerization time. One possible solution to this conundrum is that, while true isomerization occurs on the low-lying dication states much more slowly than the experimental timescale, significant large-amplitude proton motion can occur on the sub-100 fs timescale without leading to isomerization. As we will demonstrate, those dynamics can be easily misinterpreted as actual isomerization in Coulomb explosion measurements.

In the higher-lying $1\pi_u^{-1}3\sigma_g^{-1}$ states of the dication, the isomerization channel is barrierless (Supplementary Fig. 4).

However, the double valence hole in these states weakens the C–C bond. The C–C bond of the dication therefore lengthens from 1.21 to 1.37 Å, with a relaxation energy of ~2.4 eV. The weak C–C bond coupled with the large energy release implies facile dissociation and fragmentation competes with isomerization. Even if there is no isomerization, significant large amplitude motion of the hydrogen atoms would be expected. During or after the fragmentation, the fragments might rotate relative to each other. Upon Coulomb explosion, this relative rotation could masquerade as isomerization. As we will see, the experimental deuteron migration signal is well reproduced by our simulations. However, the signatures previously thought to arise from isomerization instead arise from the relative rotation of the fragments after breakup.

While the fragmentation and subsequent relative rotation of the fragments account for the major signal seen in the experiment, we did identify a sub-100 fs isomerization pathway that starts from a highly dissociative $1\pi_u^{-1}3\sigma_g^{-1}$ state and ends up with isomerized $[CH_2C]^{2+}$ on a $1\pi_u^{-2}$ state. This channel is predicted to be highly improbable, being observed only once out of 500 initial conditions and even then with a transition probability of $<5 \times 10^{-4}$ (implying an estimated probability of $<1 \times 10^{-6}$). Nevertheless, it is an interesting channel because it isomerizes through nonadiabatic transitions. Essentially, the molecule isomerizes on the $1\pi_u^{-1}3\sigma_g^{-1}$ state, where torsion is facile. A nonadiabatic transition then takes the molecule to the lower lying $1\pi_u^{-2}$ state, on the vinylidene side of the isomerization barrier. This mechanism is only possible because of the breakdown of the Born–Oppenheimer approximation.

In this work, we present a complete theoretical time-resolved picture of the ultrafast X-ray pump/X-ray probe experiment on acetylene dication dynamics. We model the dynamics of the core-ionized cation, its Auger decay, the dynamics of the dication, and the momentum distribution in the time-resolved Coulomb explosion imaging that was used to record a molecular movie for acetylene dication dynamics[5], as shown schematically in Fig. 1. Our results show that a sub 100 fs

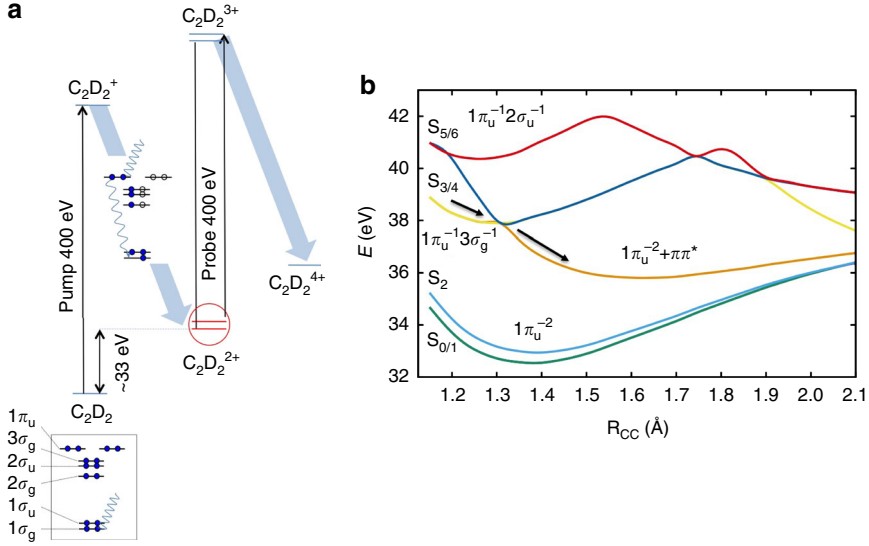

**Fig. 1** Illustration of relevant dynamical processes after core ionization of acetylene. **a** Sketch of the acetylene dication dynamics induced by X-ray photoionization and Auger decay. A first X-ray pump pulse core ionizes the neutral molecule to create the cation, which then undergoes Auger decay. A second X-ray probe pulse with a variable delay further core ionizes the dication, which promptly undergoes further Auger decay and Coulomb explosion. The momentum of the resulting fragments is measured to create the momentum map described in the text. **b** The potential curves of the singlet dicationic states are plotted in the adiabatic representation. The first 3 adiabatic states $S_0$–$S_2$ are dominated by a double hole configuration $1\pi_u^{-2}$, and the higher-lying $S_3$ and $S_4$ states have the double hole configuration $1\pi_u^{-1}3\sigma_g^{-1}$, with one electron hole in each of the $\pi$- and $\sigma$-orbitals. The *black arrows* label the barrierless fragmentation pathway on $S_{3/4}$, arising from a crossing of the diabatic states $^1\Pi_u\left(1\pi_u^{-1}3\sigma_g^{-1}\right)$ and $^1\Sigma_u\left(1\pi_u^{-2} + \pi_u \rightarrow \pi_g^*\right)$

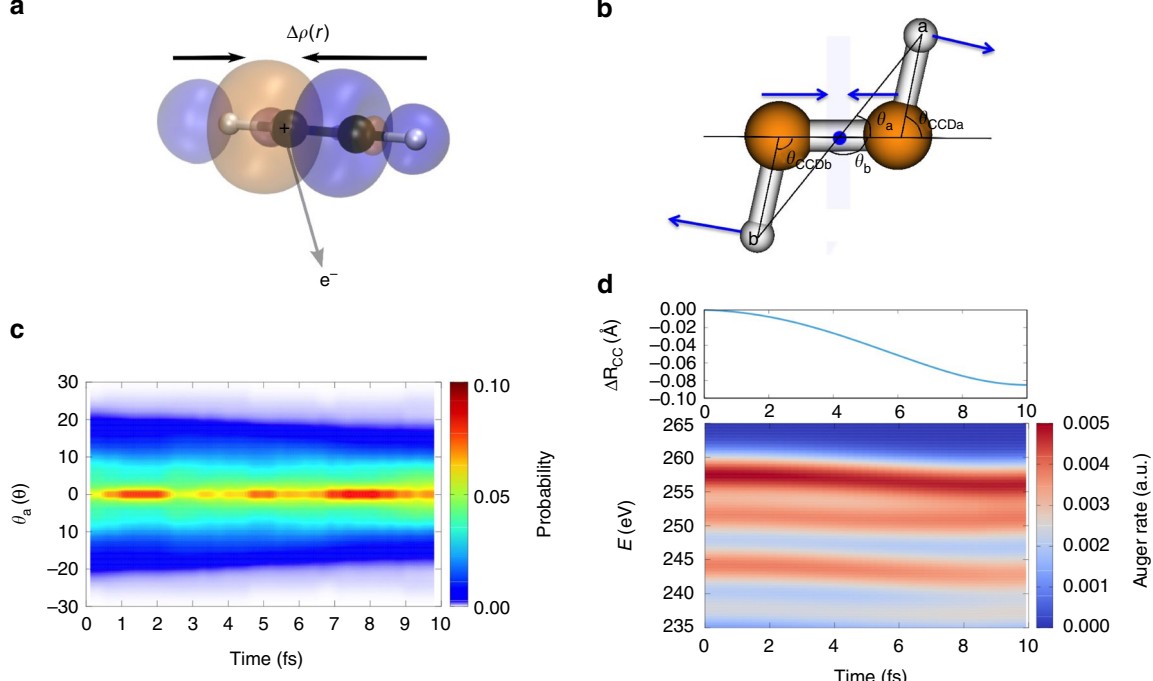

**Fig. 2** Dynamics of the core ionized acetylene cation. **a** The relaxed electron density $\rho_{-1}(r)$ after carbon $K$ edge photoionization. $\Delta\rho(r) = \rho_{-1}(r) - \tilde{\rho}_{-1}(r)$ is shown, where $\tilde{\rho}_{-1}(r)$ is the electron density of the unrelaxed core hole state after removing one electron from $C1s$ orbital. **b** Characteristic motion of cation, C–C bond contraction and CCD linearization. **c** Evolution of the CCD angle $\theta_a$ shows the cation evolves towards a narrower angular distribution. **d** The time-resolved Auger spectra from a representative trajectory and the evolution of C–C bond length and kinetic energy of Auger electrons. The contracting C–C bond results in a *red shift* of Auger electron energies. The Auger spectra are broadened by a Lorentzian corresponding to a core-hole lifetime of 8 fs

isomerization in the low-lying electronic states of the acetylene dication is unlikely.

## Results

**Dynamics of the core ionized acetylene cation.** In our simulation, we follow the experiment shown schematically in Fig. 1(a), and use $C_2D_2^+$ rather than $C_2H_2^+$. Deuterated acetylene was employed in the experiment to eliminate potential background sources of protons from water and other contaminants[5]. Although deuterated acetylene should have the same electronic structure except for negligible second-order Born–Huang coupling, the particle velocity and the rate of tunneling through the barrier are expected to be slower in $C_2D_2^+$ compared with $C_2H_2^+$. The experiment was interpreted to show strong signatures of a vinylidene-like channel already in the first 12 fs following core ionization from Coulomb explosion momentum mapping (CEMM)[5], implying that the X-ray induced dynamics starts immediately after photoionization of the carbon $K$ shell. We model the dynamics on the core-hole state prior to the Auger decay that yields the dication. The potential energy surface of the core ionized cation on the $^2\Sigma_{g/u}$ state is calculated (during the dynamics) using a $\Delta$SCF scheme with the maximum overlap method (details in Supplementary Note 1)[20]. For modeling the core hole state, we adopt a localized picture with the core hole localized on one of the carbon atoms (i.e., breaking the gerade/ungerade symmetry). As shown in Fig. 2a, the dynamical screening of the core hole through shake-up processes enhances the valence electron density in the C–C bond region. The electron reorganization strengthens the bonding along the molecular axis, and hardens the angular potential for the deuterons along the $\theta_{CCD}$ coordinate (as discussed in Supplementary Note 1). Our ab initio molecular dynamics simulations of core ionized acetylene cation start from initial conditions (positions and momenta of the atoms) sampled from the vibrational ground-state harmonic

Wigner distribution of neutral acetylene, as calculated with second-order perturbation theory (MP2) in the 6-31G* basis set.

Figure 2c, d presents the evolution of the angular distribution of the deuterons and the C–C bond distance ($R_{C-C}$) after core-hole ionization by the X-ray pump pulse (on the cationic $^2\Sigma$ state). The nuclear dynamics after core ionization is rather limited and mainly characterized by a decrease in $R_{C-C}$ and narrowing of the CCD bending angle distribution. Along the core-hole ionized trajectories, we calculate the instantaneous Auger spectra (Fig. 2d and Supplementary Note 2)[21]. The energy of the emitted Auger electron is lowered with time because $R_{C-C}$ decreases on the cationic $^2\Sigma$ state (increasing the Coulomb repulsion of the two holes in the dication that results from Auger decay). Thus, the energy difference between the core ionized state and the repulsive dicationic final state decreases as $R_{C-C}$ decreases. This effect leads to a redshifted Auger spectrum as the dynamics proceeds. Given the time-resolved Auger spectrum along each of the cationic trajectories, we populate the first five singlet and the first four triplet dicationic states with probabilities derived from the decay rate of the $K$ shell core hole state $|K\rangle$ to different dicationic states $|LL\rangle$ with double holes in the $L$-shell as $P_{KLL_i} = \rho_K(t)\Gamma_{KLL_i}(t)\Delta t$, where $\Gamma_{KLL_i}(t)$ is the Auger rate at a delay time of $t$ after $C1s$ core ionization and $\rho_K(t)$ is the population of the core ionized state determined by the kinetic rate equation $\dot{\rho}_K(t) = -\rho_K(t)\Gamma(t)$, $\rho_K(0) = 1$, where $\Gamma(t) = \sum_{|KLL_i\rangle} \Gamma_{KLL_i}(t)$ is the total Auger decay rate. We assume that the initial positions and momenta of the dication are inherited from the parent cation at the time of Auger decay and a recoil momentum from Auger electron is added on the carbon atom subject to primary core-ionization. In the following discussion, we focus on the lowest five singlet and four triplet dicationic states. Higher-lying states are not relevant because they are directly dissociative. Thus, population of these states will not lead to isomerization, but rather immediate fragmentation, which is too fast to allow deuteron migration

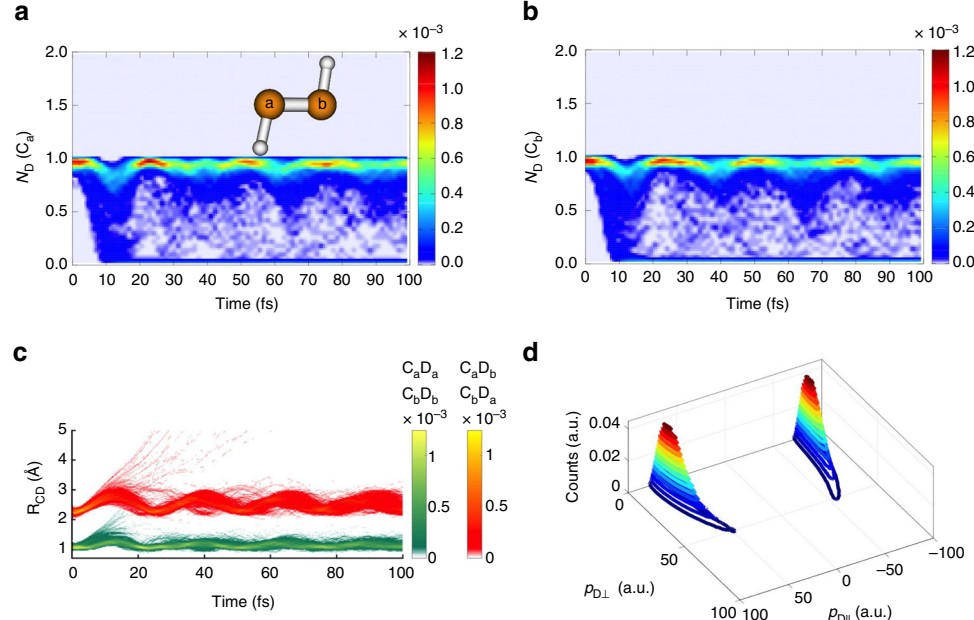

**Fig. 3** Dynamics of acetylene dication after Auger decay. Deuteron coordination number ($N_D$) of the two carbon atoms **a** $C_a$ and **b** $C_b$. See main text for definition of the deuteron coordination number. **c** The C–D distances ($R_{CD}$) for the initially bonded $C_a$–$D_a$ and $C_b$–$D_b$ atoms (*green color*), and for the initially nonbonded $C_a$–$D_b$ and $C_b$–$D_a$ atoms (*orange color*). **d** Effective momentum distribution produced from non-dissociating trajectories, assuming no remnant rotation between momentum vectors during the Coulomb explosion. $p_{D\perp}$ and $p_{D\parallel}$ are the components of deuteron momenta that are perpendicular and parallel to the C–C axis, respectively

across the C–C bond. These higher-lying states will therefore result in either symmetric breakup ($CH^+ + CH^+$) or deprotonation ($C_2H^+ + H^+$) channels. Additionally, the Auger decay rates into triplet states that support deuteron migration are one order of magnitude lower than those for the dominant singlet states. Thus, it is primarily the lowest five singlet states that are important to the question of isomerization in the X-ray pump experiments we discussed in the introduction[5, 12, 13], and we focus on these from here on. Generally the acetylene cation decays into $C_2D_2^{2+}$ dication with an Auger lifetime of ~8 fs.

**Acetylene dication dynamics**. We now focus on the dication dynamics initiated after Auger decay, and explore the possibility of isomerization within 100 fs. Molecular dynamics on the dicationic states are described with the *ab initio* multiple spawning (AIMS) method[22]. At each time step, the electronic structure is solved using a state-averaged complete active space self-consistent field (CASSCF) wavefunction[23] with eight active electrons in eight orbitals using the 6-31G* basis set. The corresponding potential energies, gradients and non-Born–Oppenheimer couplings necessary for the classical evolution of the nuclear coordinates and solution of the nuclear Schrödinger equation in the time-evolving basis set of Gaussian wavepackets centered on classical trajectories are computed on the fly[24–27]. The coupled electron-nuclear dynamics of $C_2D_2^{2+}$ is simulated with the AIMS method[22] (Supplementary Note 3), which solves the electronic and nuclear Schrödinger equations simultaneously using a basis set of travelling Gaussian wavepackets for the nuclear wave function and determining the electronic structure as needed with the CASSCF method[28, 29]. Although nonadiabatic crossing effects can be described by AIMS, we found that the short time (sub-100 fs) dynamics of the dication after Auger decay is almost entirely adiabatic and the population of each state can be considered as constant on this timescale.

As shown in Supplementary Fig. 6a, trans-bending of the dication is energetically highly disfavored on the low-lying $1\pi_u^{-2}$

states. In contrast, trans-bending up to ~60° is possible on the $1\pi_u^{-1}3\sigma_g^{-1}$ states. The different behavior of these electronic states can be understood from the nature of the bonding. The σ-bond is located along the C–C axis, and favors a linear C–C–D structure. Removing an electron from the bonding σ-orbital weakens the σ-bond, leading to more freedom for the deuterons to bend such that the $\theta_{CCD}$ angle is increased. Energetically, large amplitude bending (which could lead to isomerization) of the deuterons is possible on the $1\pi_u^{-1}3\sigma_g^{-1}$ states, and it is conceivable that this could take place on the sub-100 fs timescale. However, in these states, it is also energetically favorable to break the C–C bond (Supplementary Fig. 6b) and dissociation could compete with isomerization. For the lowest three $1\pi_u^{-2}$ states, the large isomerization barrier (≈2 eV) suggests that isomerization on these states is highly unlikely to complete on sub-100 fs timescale. Thus, for all the low-lying singlet states, we might expect ultrafast isomerization to be a rare channel. On the $1\pi_u^{-2}$ states, the barrier is too high and on the $1\pi_u^{-1}3\sigma_g^{-1}$ states, the symmetric breakup or deprotonation channels might be more likely. As shown in Fig. 3, the simulations (corresponding to an ensemble of 500 trajectory basis functions for singlet and triplet states) clearly do not observe any isomerization in the first 100 fs of the dication dynamics (recall the preparation of the dication follows from direct modeling of the X-ray pump and subsequent Auger decay). Figure 3a, b depicts the time evolution of the probability distribution for the deuteron-coordination number $N_D(C_i)$ of the left and right carbon atoms, defined as ref. [30]:

$$N_D(C_i) = \sum_j S(|r_{D_j} - r_{C_i}|), \quad i, j = a, b \qquad (1)$$

where

$$S(r) = 1/(\exp[\kappa(r - r_c)] + 1) \qquad (2)$$

and we choose $r_c = 1.4$ Å and $\kappa^{-1} = 0.1$ Å. The coordination number $n_C$ provides a smoothed count of the number of deuterons within bonding distance of each carbon atom. In no

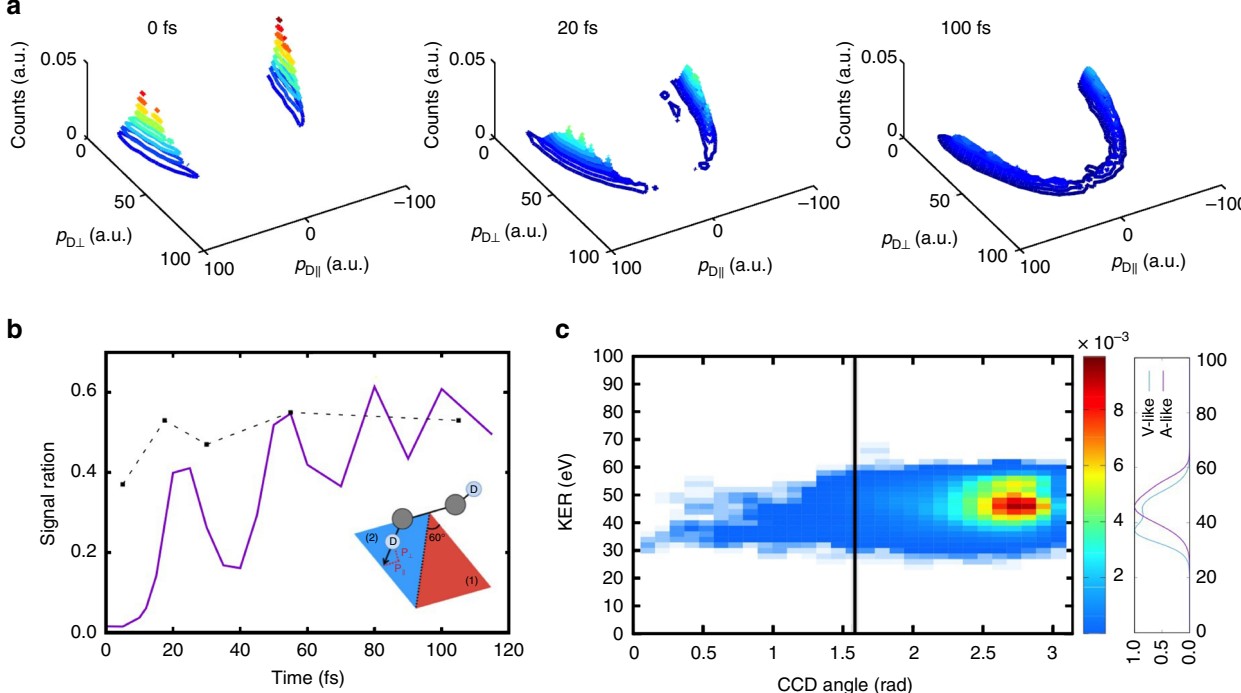

**Fig. 4** Simulated Coulomb explosion momentum mapping signal. **a** Temporal evolution of the deuteron momenta from 0 to 100 fs (axes correspond to directions parallel and perpendicular to the C–C axis) with an assumed instrumental broadening of $3.8 \times 10^{-22}$ kg m s$^{-1}$. **b** Ratio of signal from *red-shaded* region where $0° \leq \tan^{-1}(p_{\parallel}/p_{\perp}) < 60°$ and *blue-shaded* region, where $60° \leq \tan^{-1}(p_{\parallel}/p_{\perp}) < 120°$. *Solid line* is from simulations, which is compared with the experiments (*dashed line*). **c** The total kinetic energy release (KER) and angular distribution of $C+/C+/D+/D+$ coincidences integrated over all time delays. The CCD angle is defined as $\tilde{\theta} = \cos^{-1}\left(\frac{\text{sgn}\left[(p_{C_a} \cdot p_{D_2})\right]\left((p_{C_a} - p_{C_b}) \cdot p_{D_1}\right)}{|p_{C_a} - p_{C_b}||p_{D_1}|}\right)$, as the angle a deuteron momentum makes with the effective C–C axis[5]

case does this coordination number exceed one, which is a clear sign that no isomerization occurs on this timescale. Isomerization would entail one of the carbon atoms having a coordination number near two, while the other would have a coordination number near zero. For example, the CASSCF-optimized geometry of $CCD_2^{2+}$ on the $^1\Delta_g$ electronic state yields coordination numbers $N_D(C_{left})$ and $N_D(C_{right})$ of 0.0 and 1.9, respectively. Figure 3c shows the four C-D distances as a function of time, grouped such that the longer two such distances (at time $t = 0$) are colored in red and the shorter two are colored in green. Again, it is quite clear that there is no switching of a deuteron between carbon atoms. This would appear to be inconsistent with the experiments that showed apparent evidence of ultrafast isomerization. In order to resolve this conundrum, we also simulated the Coulomb explosion process so that we could compare directly to the experimental observables. It is worth noting that the experimental analysis of the Coulomb explosion data often assumes that the explosion is so fast that all rotation stops immediately. Under this approximation, the momentum map, which would be expected from the experiment if there was no evolution on the dication state (neither isomerization nor dissociation), is shown in Fig. 3d (details in Supplementary Note 3). Although the sudden approximation is clearly not strictly valid, as has been observed previously[18], it is necessary to connect the measured CEMM data to a unique structure.

Following the recent experiment[5], we simulated the CEMM image that was used to measure the nuclear motion. At each delay time and for each of the trajectory basis functions from the dication dynamics, we placed point charges (with appropriate masses) at the locations of each of the nuclei and propagated classically to simulate the experiment directly. Figure 4a shows the time evolution of the momentum map, which can be

compared to Fig. 3 of ref. [5]. Note that the region with large momentum perpendicular to the C–C axis and small momentum along the C–C axis fills in as the time delay increases. This was the primary observation in ref. [5], leading to the conclusion that isomerization was occurring. In order to make the comparison more clear, we also plot the kinetic energy release distribution as a function of $\angle$CCD ($\theta_{CCD}$) in Fig. 4c, again following the experimental data analysis. The natural assumption here is that population with $\angle$CCD less than $\pi/2$ are vinylidene-like, i.e., isomerized. Figure 4c can be compared directly with Fig. 2 of ref. [5]. and (as in the experiment) shows considerable signal in the vinylidene like region (to the left of the black line in Fig. 4c). This is in spite of the fact that there is no isomerization in the simulation data. Furthermore, the redshift in KER for the V-like channel observed in the experiment is also seen in the simulations (this redshift is due to kinetic energy loss when there is transbending of the deuteron). The V-like signal with low KER originates from higher energy trajectories with significant proton motion. Due to angular momentum conservation, rotation continues during Coulomb explosion for both the $C_2D^+ + D^+$ deprotonation and $CD^+ + CD^+$ symmetric breakup channels. The Coulomb explosion of the rotating fragments leads to V-like signals in the momentum mapping (Supplementary Fig. 10 and Supplementary Note 3). Thus, while these channels do contain significant deuteron motion, they do not result in isomerization and would not break up into $C^+/CD_2^+$ had they not been Coulomb exploded. In Supplementary Fig. 7, we show the dynamics of C–C axis rotation, which was used as a clock in an earlier acetylene dication experiment that was also interpreted to support ultrafast isomerization[12]. Due to significant C–C bond elongation, its rotation decelerates. Supplementary Fig. 7 shows that the C–C axis rotation depends linearly on time for only the

first 30 fs. After that time, the C–C axis rotation is nearly time-independent for the 100 fs of our simulations. This implies that the clock is only accurate up to ≈30 fs and the true upper bound for the isomerization reaction is not 60 fs but rather some longer timescale (which is difficult to quantify without an improved model to calibrate the clock).

Although the AIMS simulations do not support the isomerization channel, isomerization could take place with very low probability. Because the non-Born–Oppenheimer coupling allows transitions between electronic states induced by nuclear motion[8, 24, 27, 31–33], the dication can undergo partial proton migration (large $\theta_{CCD}$ angle) on the $1\pi_u^{-1}3\sigma_g^{-1}$ states, which have a soft (nearly flat) angular potential, and then decay downwards via an electronic transition to the $1\pi_u^{-2}$ states where it might complete the isomerization. This nonadiabatically-assisted isomerization mechanism depends on the electronic transition occurring after the isomerization barrier on the lower $1\pi_u^{-2}$ states. Such a trajectory is shown in Supplementary Fig. 6c with the trans-bending motion to $\theta_{CCD}$ angle of ~80° within 60 fs on the state $S_4$ and $S_3$ with $1\pi_u^{-1}3\sigma_g^{-1}$ or $1\pi_u^{-2} + \pi_u \rightarrow \pi_g^*$ character, and it completes isomerization on $S_2$ at ~90 fs. However, nearly all trajectory basis functions with high trans-bending angles on states $S_3$ and $S_4$ do not follow this path but instead fragment along the C–C bond because of the high vibrational energy in the C-C stretch and the barrierless character of the potential along this coordinate (Supplementary Fig. 6b). Using a simplified model (Supplementary Note 3), we can estimate the branching ratio of the nonadiabatically-assisted isomerization channel and the kinematically favored C–C symmetric fragmentation channel to be ~$1 \times 10^{-4}$. This estimate can be compared to the estimate of $1 \times 10^{-6}$ from the AIMS simulations.

We further point out that isomerization is possible from the satellite state $^1\Sigma_u$ ($S_7$) with 3-hole-1-particle electronic character $1\pi_u^{-2} + \pi_u \rightarrow \pi_g^*$. The $^1\Sigma_u$ state is accessed from the shake-up state $(C_{1s})^{-1} + \pi_u \rightarrow \pi_g^*$ in photoionization with Auger energy of 256.8 eV. It crosses the $^1\Pi_u$ and $^1\Pi_g$ states, and could switch to these states that support ultrafast isomerization. Using the sudden approximation, we can estimate the $K$-shell photoionization cross section as $\sigma \sim \left| \left\langle \psi_{M^+}^{N-1} | \hat{a} | \psi_{M^0}^N \right\rangle \right|^2$ and determine the ratio of the shake-up state $(C_{1s})^{-1} + \pi_u - \pi_g^*$ to the $(C_{1s})^{-1}$ state to be ~0.03. Due to their extremely low probability, these channels alone are insufficient to explain the abundant vinylidene-like signals found in the experiment.

To further investigate the possibility of an isomerization pathway, we take advantage of the fact that Coulomb explosion imaging is sensitive to the actual C–C bond length at the time of tetracation generation. An unbroken C–C bond is expected to result in V-like signal with higher KER due to larger Coulomb potential of short C–C distance, from which we can remove the symmetric breakup channel with significant hydrogen migration. We specifically look at the momentum difference of coincident carbon ions $p_{C^+C^+} = |p_{C_a^+} - p_{C_b^+}|$. We could not identify any signal in the experimental data set[5] beyond the simulated CEMM signals from non-isomerized trajectories at sufficient confidence level, as detailed in Supplementary Note 3. We can thus conclude with high confidence that isomerization is not occurring with any significant probability in acetylene dication prepared by Auger decay after X-ray core ionization[5].

As a final note, we also observe the signature of vibrational coherence in the bending motion of the acetylene dication, as seen in Fig. 4b. Since the vibrational motion of the acetylene dication is synchronized by the X-ray pump pulse when the vibrational frequency is suddenly changed (by ionization), vibrational coherence can be expected to occur for 100 fs before dephasing. This is also known as a squeezed vibrational

state[5, 34], analogous to the squeezed coherent state of photons. The vibrational coherence manifests itself in the ratio of deuteron with large bending angles (in region (1), Fig. 4b) and in the vicinity of carbon atom (in region (2)), with a period of ~27 fs, which is half of the period of trans-bending motion and is consistent with a squeezed vibrational state that gives collective vibrational amplitude proportional to $\left[ \left(1 + (\omega_0/\omega_1)^2\right) + \left(1 - (\omega_1/\omega_0)^2\right) \cos(2\omega_1 t) \right]^{1/2}$, where $\omega_0$ and $\omega_1$ are the vibrational frequencies before and after the pump pulse (details in Supplementary Note 3).

Our study resolves the long-standing controversy between experiment and theory concerning the mechanism of the purported sub-100 fs isomerization of acetylene dication. We conclude that in fact what appeared as ultrafast isomerization in previous experiments is actually significant proton migration on the ground state, or on excited states that then decays into symmetric breakup. Isomerization, which requires a stable C–C bond, can only occur in the low-lying states of the dication if the molecules have enough internal energy (and time) to overcome the isomerization barrier. This mechanism is infeasible on the sub-100 fs timescale of the pump-probe experiment modeled here. Enough energy may be available for isomerization after nonadiabatic internal conversion from high-lying dicationic states towards the low energetic states. However, in this case, direct symmetric fragmentation dominates overwhelmingly.

Our work calls for cautious interpretation of the widely used CEMM method when resolving the transient geometry of molecular motion on femtosecond timescale. On the other hand, it also highlights CEMM's ability to resolve the ultrafast dynamics of momentum dispersion. Even when no significant geometric variation takes place, CEMM reveals the rich dynamics of the momentum distribution that changes substantially on the femtosecond timescale. With complementary transient geometry information from single molecule diffraction, which is enabled by X-ray free electron lasers or relativistic electrons[35, 36], we could form a complete picture of molecular dynamics in the entire phase space, including both position and momentum. Such a time-resolved diffraction study was recently reported[37] and we expect that the combination of the simulations reported here, the previous CEMM measurements, and time-resolved diffraction will give a complete picture of the femtosecond dynamics of acetylene dication.

**Data availability**. The data sets generated during and/or analyzed during the current study are available from the corresponding author on request.

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

## Acknowledgements

This work was supported by the AMOS program within the Chemical Sciences, Geosciences and Biosciences Division of the Office of Basic Energy Sciences, Office of Science, US Department of Energy and the Hamburg Center for Ultrafast Imaging. N.M. acknowledges partial financial support from the Czech Ministry of Education (Grants LG15013 and LM2015083). Z.L. thanks the Volkswagen Foundation for support through a Peter Paul Ewald postdoctoral fellowship. Z.L. also thanks Lee-Ping Wang, Koudai Toyota, Sang-Kil Son, Robin Santra, Daniel Haxton, Daniel Neumark, Mohamed El-Amine Madjet, Kota Hanasaki, Victor Kimberg and Yajiang Hao for stimulating discussions.

## Author contributions

T.M. and Z.L.: Conceived the concept and methods. T.M., Z.L., L.I. B.C., J.S., N.M., S.P., and O.V.: Conducted the theoretical investigation. C.L.-S., J.C., T.O., and P.B.: Conducted the experimental data analysis. All authors analyzed the results and reviewed the manuscript.

## Additional information

**Competing interests:** The authors declare no competing financial interests.

