## [Peer review file · Nature Communications]

Reviewers' comments:

Reviewer #1 (Remarks to the Author):

The authors present a theoretical study on acetylene-vinylidene isomerization in $C_2D_2^{2+}$ after the inner-core $C1s$ ionization and the Auger decay of deuterated acetylene. The theoretical calculation shows the Auger decay populates numerous electronic states in the dication and those with the $1\pi_u^{-2}$ and $1\pi_u^{-1} 3\sigma_g^{-1}$ configurations are chosen for further discussion. Molecular dynamics calculations were carried out using the ab initio multiple spawning (AIMS) method to study the isomerization process. They conclude that the isomerization is unlikely to occur in the timescale (~ 100 fs) suggested by the Coulomb explosion imaging experiments (ref.5). The three singlet states in the $1\pi_u^{-2}$ configuration have high isomerization barriers (~ 2 eV). As for the remaining $1\pi_u^{-1} 3\sigma_g^{-1}$ states, they argue that the isomerization is quite improbable because of the competition with the C-C bond breaking. By simulating the fragment momentum produced by the probe pulse, the authors show that the experimental results were 'misinterpreted' in ref.5. The paper reports a complete study on the complicated molecular dynamics associated with X-ray core ionization, which will be highly appreciated by the broad audience. The paper deserves publication in Nat Comm, but there are several issues that should be addressed before publication.

- 1) As described above, the present study was carried out on the isomerization associated with the X-ray core ionization of acetylene. In this respect, the title should be more specific. I suggest "Ultrafast Isomerization in Acetylene Dication by Carbon K-shell Ionization: ..."
- 2) The exclusion of the isomerization from the $1\pi_u^{-1} 3\sigma_g^{-1}$ state seems to be an important point to support the conclusion of the paper. In this respect, the derivation of the ratio (2.5×10^{-4}) to the C-C breaking channel should be explained more in detail. For example, what are m_i , V in the equation in Supplemental Material (and their values)? In addition, the authors assume that the energy is randomly distributed in the five degrees of vibrational freedom. The time scale (100 fs) is too short to complete intramolecular vibrational energy redistribution (IVR).
- 3) The bond order in Figs.3a and 3b is below 1 even at $t = 0$ fs. Why? It might be possible that the bond order can be below 1 even in the "vinylidene" form by the definition of the bond order or the choice of the relevant parameters in Eq.(1). Comparison with the bond order calculated for excited vinylidene dication should be useful to support their discussion.
- 4) The authors attributed the misinterpretation of the Coulomb explosion imaging data in ref.5 to the rotation of the fragment ions prior to the probe laser pulse. I agree with the authors, but the effect of fragment rotation on CEI data has been already discussed before (for example, in ref. 18), which should be commented (briefly) in the main text.
- 5) Line 168: 'Sec. IIID' should be 'Sec IIIB'.

Reviewer #2 (Remarks to the Author):

The authors report on a theoretical analysis of the dynamics in the acetylene dication produced by Auger decay after X-ray photoionization of the carbon atom K shell. The interpretation relies on the state-of-the-art dynamics simulations on the singlet dicationic states using the ab initio multiple spawning method. The triplet dicationic states are not taken into account as according to the authors the Auger decay rates into these states is negligible. The Coulomb explosion momentum map (CEMM) is simulated by placing point charges at the location of the nuclei at each time delay and for each of the trajectory basis functions of the dication dynamics, which are then propagated classically. In the simulations no significant isomerization occurs on the experimentally reported time scale, nevertheless similar patterns are observed in the CEMM as in the experiment, which had been interpreted as signatures for isomerization acetylene to vinylidene in the dication.

I find the paper very interesting and certainly important to the field. But I am not sure whether it suitable for nature comm. as the experiments have already been published. Besides that some points should be addressed or clarified.

The trajectory calculations are performed on CASSCF level of theory. A benchmark of the barrier height towards isomerization in the relevant singlet states, which is crucial for the dynamics, should be given on CASPT2 or CC2 level of theory. This will give some scale, how to interpret the dynamical calculations.

Did the authors check that the Eckart conditions between different trajectories are fulfilled? This is important to rule out any artificial rotations for the subsequent simulation of the Coulomb explosion.

The labeling of nearly all the figures is much too small. They are not readable in the print out.

I also think the title is much too general - though admittedly fancy. Experiments by i.e. the Kling group (ref. 10) and by the Kitzler group reported isomerization of acetylene in the dicationic state. In their experiments the relevant dynamics occurred in the lowest triplet Pi state, which has a much lower barrier and a higher lying transition state structure at the geometry of the neutral acetylene. Thus its barrier is easier to be overcome.

It would also be interesting to discuss the possibility of spin orbit coupling from the relevant singlet states to the triplet states, especially the triplet Pi state. This might open a more effective route to isomerization.

Reviewer #3 (Remarks to the Author):

The manuscript analyzes possible mechanisms of isomerization of acetylene dication to vinylidene dication by ultrafast H migration, which was concluded to occur within 100 fs based on recent Coulomb explosion experiments. This experimental observation is at odds with theoretical calculations of the potential energy surfaces and rate constants for isomerization of acetylene dication in low lying electronic states, which showed that the H migration should be much slower. The authors of the present work succeeded in resolving this mystery. They computed the dynamics of acetylene dication produced by Auger decay following X-ray photoionization of the carbon atom K shell; these nuclear dynamics calculations involved several electronic states of the dication. The results show that the experimental observation can be assigned in terms of decomposition of the dication into two fragments ($\text{CH}^+ + \text{CH}^+$ or $\text{C}_2\text{H}^+ + \text{H}^+$) involving rotation of the departing fragments relative one another, rather than in terms of the H migration. The fast H migration channel involving a non-adiabatic transition between different electronic states has been also found but the authors show that probability of this process is rather low and it cannot be responsible for the experimental observation.

The calculations have been carried out at a state-of-the-art level and the results are convincing. There are two important contributions of this work to the field: 1) the seeming contradiction between theory and experiment on the rate of H migration in acetylene dication, which in fact was present in the literature for the last two decades, has been resolved and 2) the authors demonstrated a theoretical approach for better interpretation of experimental data derived from the widely used Coulomb momentum imaging method.

On these grounds, I believe that this manuscript merits publication in Nature Communications.

I have one minor comment for the authors:

References 5 and 23 are identical.

We thank the three Referees for the positive evaluation of our work and their constructive comments that helped us to further improve the manuscript. Below we present our detailed replies to the Referees' comments. New text is indicated in blue and deleted text is indicated in red with strikethrough.

Referee 1:

The authors present a theoretical study on acetylene-vinylidene isomerization in C2D22+ after the inner-core C1s ionization and the Auger decay of deuterated acetylene. The theoretical calculation shows the Auger decay populates numerous electronic states in the dication and those with the $1\pi_u^{-2}$ and $1\pi_u^{-1}3\sigma_g^{-1}$ configurations are chosen for further discussion. Molecular dynamics calculations were carried out using the ab initio multiple spawning (AIMS) method to study the isomerization process. They conclude that the isomerization is unlikely to occur in the timescale (~ 100 fs) suggested by the Coulomb explosion imaging experiments (ref.5). The three singlet states in the $1\pi_u^{-2}$ configuration have high isomerization barriers (~ 2 eV). As for the remaining $1\pi_u^{-1}3\sigma_g^{-1}$ states, they argue that the isomerization is quite improbable because of the competition with the C-C bond breaking. By simulating the fragment momentum produced by the probe pulse, the authors show that the experimental results were 'misinterpreted' in ref.5. The paper reports a complete study on the complicated molecular dynamics associated with X-ray core ionization, which will be highly appreciated by the broad audience. The paper deserves publication in Nat Comm, but there are several issues that should be addressed before publication.

1) As described above, the present study was carried out on the isomerization associated with the X-ray core ionization of acetylene. In this respect, the title should be more specific. I suggest "Ultrafast Isomerization in Acetylene Dication by Carbon K-shell Ionization: ..."

We thank the reviewer for her positive assessment of our work. We agree that our paper addresses a specific X-ray core ionization and Auger decay process that mainly populates the singlet $1\pi_u^{-2}$ and $1\pi_u^{-1}3\sigma_g^{-1}$ double-hole states of acetylene dication. We have changed the title to "Ultrafast Isomerization in Acetylene Dication **After Carbon K-shell Ionization: To Be or Not to Be**" in the revised manuscript.

2) The exclusion of the isomerization from the $1\pi_u^{-1}3\sigma_g^{-1}$ state seems to be an important point to support the conclusion of the paper. In this respect, the derivation of the ratio (2.5×10^{-4}) to the C-C breaking channel should be explained more in detail. For example, what are m_i , V in the equation in Supplemental Material (and their values)? In addition, the authors assume that the energy is randomly distributed in the five degrees of vibrational freedom. The time scale (100 fs) is too short to complete intramolecular vibrational energy redistribution (IVR).

The most compelling evidence against this channel are the combined facts that 1) we only observe this channel approached once out of 500 initial conditions and even then, the probability of the required nonadiabatic transition is less than 1×10^{-4} and 2) our simulations reproduce the bulk of the observed experimental signal in the absence of this channel. We have clarified this in the main text:

Page 3: This channel is predicted to be quite improbable: ~~2.5×10^{-4} times less likely than~~

the symmetric breakup channel.20, being observed only once out of 500 initial conditions and even then with a transition probability of less than 5×10^{-4} (implying an estimated probability of less than 1×10^{-6}).

The additional information in the supporting material was intended as a further justification and not as the primary evidence. Nevertheless, we have clarified the analysis according to the reviewer's comments. In the supplemental material, m_i are the reduced masses of the corresponding modes. V is the quantization volume, which cancels out in the ratio of the C-C breaking and isomerization (i.e. trans-bending) channels. The reduced masses are taken as are taken as 7 amu and 2.592 amu for the C-C breaking and trans-bending modes from spectroscopic data [M. Hochlaf *et al.*, J. Chem. Phys. 132, 194301 (2010)]. We have added the definition of the variables and the values of the reduced masses in the supplemental material of the revised manuscript. We agree with the referee concerning the timescale for IVR and have noted that in the supporting material (but as stated above, the argument in the supporting material is only provided as a rationalization and not as primary evidence for exclusion of isomerization on the $1\pi u-13\sigma g-1$ state.)

3) The bond order in Figs.3a and 3b is below 1 even at $t = 0$ fs. Why? It might be possible that the bond order can be below 1 even in the "vinylidene" form by the definition of the bond order or the choice of the relevant parameters in Eq.(1). Comparison with the bond order calculated for excited vinylidene dication should be useful to support their discussion.

It is important to note that the quantity plotted in Figures 3a/3b is not a bond order (which would be calculated from electronic structure information) but rather a coordination number (defined purely in terms of geometric quantities, i.e. atom-atom distances). We have tried to clarify this by specifying that this is a deuteron coordination number.

Page 7: Figures 3a and 3b depict the time evolution of the probability distribution for the deuteron-coordination number $ND(C_i)$ of the left and right carbon atoms,

As a technical point, the calculation of the deuteron-coordination number (described in Eqs. 1 and 2 of the main text, which is now indicated explicitly in the figure caption) involves convolution of the deuteron density with a smooth function $S(r)$ around the carbon atoms. This definition follows Ref. 30. Because of the smoothness, even at the equilibrium geometry, the coordination number is always slightly below one since the function $S(r)$ is not strictly zero at equilibrium bond length as shown schematically below (for our original choice of $\kappa=3.3\text{\AA}^{-1}$):

In order to make Figure 3 clearer, we have decreased the width of the transition region where the coordination number goes from 1 to 0 (by increasing κ from 3.3\AA^{-1} to 10\AA^{-1} in Eq. 2). As can be seen in the revised Figure 3a/3b, the coordination number is now closer to 1 at the initial time. We agree with the reviewer that some information about the values of these coordination numbers in the isomerized species would also be useful. Thus, we have modified the text:

Page 8: ...and we choose $r_c=1.4\text{\AA}$ and $\kappa^{-1}=0.1\text{\AA}$. The coordination number n_C provides a smoothed counts of the number of deuterons within bonding distance of each carbon atom. In no case does this coordination number exceed one, which is a clear sign that no isomerization occurs on this timescale. Isomerization would entail one of the carbon atoms having a coordination number of near two, while the other would have a coordination number of near zero. For example, the CASSCF-optimized geometry of CCD22+ on the $1\Delta_g$ electronic state yields coordination numbers ND(Cleft) and ND(Cright) of 0.0 and 1.9, respectively.

4) The authors attributed the misinterpretation of the Coulomb explosion imaging data in ref.5 to the rotation of the fragment ions prior to the probe laser pulse. I agree with the authors, but the effect of fragment rotation on CEI data has been already discussed before (for example, in ref. 18), which should be commented (briefly) in the main text.

We agree and have added this reference in the text:

Page 8: Although the sudden approximation is clearly not strictly valid, as has been observed previously,18 ...

5) Line 168: 'Sec. IIID' should be 'Sec IIIB'.

We thank the referee for pointing out this typographical error, which we have corrected.

Reviewer 2:

The authors report on a theoretical analysis of the dynamics in the acetylene dication

produced by Auger decay after X-ray photoionization of the carbon atom K shell. The interpretation relies on the state-of-the-art dynamics simulations on the singlet dicationic states using the *ab initio* multiple spawning method. The triplet dicationic states are not taken into account as according to the authors the Auger decay rates into these states is negligible. The Coulomb explosion momentum map (CEMM) is simulated by placing point charges at the location of the nuclei at each time delay and for each of the trajectory basis functions of the dication dynamics, which are then propagated classically. In the simulations no significant isomerization occurs on the experimentally reported time scale, nevertheless similar patterns are observed in the CEMM as in the experiment, which had been interpreted as signatures for isomerization acetylene to vinylidene in the dication.

I find the paper very interesting and certainly important to the field. But I am not sure whether it suitable for nature comm. as the experiments have already been published.

The fact that the experiments are already published seems to us to STRENGTHEN the case for publication in Nature Comm. Previous experimental and theoretical work led to the paradox in the literature and our work solves this paradox in an elegant way. Since the Coulomb explosion technique is widely used and a natural method for state-of-the-art femtosecond X-Ray lasers, we feel that it is especially important that workers in the field are aware of this example that highlights its limitations. Primarily, our work shows that one should carry out detailed simulations in order to properly interpret results from CEMM. We believe this work is of importance for a broad readership including both theoreticians and experimentalists in ultrafast molecular science, including those involved in science at femtosecond X-Ray free electron laser facilities.

Besides that some points should be addressed or clarified.

The trajectory calculations are performed on CASSCF level of theory. A benchmark of the barrier height towards isomerization in the relevant singlet states, which is crucial for the dynamics, should be given on CASPT2 or CC2 level of theory. This will give some scale, how to interpret the dynamical calculations.

We agree with the referee on the benchmark with CASPT2 theory. We have added the results at the MS8-CASPT2(8,10) level in Figures S5e and S5f of the supplemental materials in the revised manuscript. These can be compared to Figures S5c and S5d, showing the reliability of the potentials obtained using SA8-CASSCF(8,8).

Did the authors check that the Eckart conditions between different trajectories are fulfilled? This is important to rule out any artificial rotations for the subsequent simulation of the Coulomb explosion.

We do not agree with the referee regarding the Eckart conditions. What is detected in the experiment and also calculated from the *ab initio* molecular dynamics trajectories are the *relative* momenta of the ejected fragments, for a set of randomly oriented molecules. Note that the first pump pulse induces core ionization, and it is well known that the propensity for such core ionization is nearly isotropic. Thus, there is practically no alignment of the core-excited molecules. Because there is no alignment before or because of the excitation pulse, we are not relating the momenta to laboratory axes with respect to which a molecular frame (via e.g. Eckart conditions) would have to be defined. Instead,

the measurement is directly related to the molecular frame of the system at the moment of ionization. Thus, the different molecules in the experiment do not satisfy the Eckart conditions and neither should the initial conditions in the simulation.

The labeling of nearly all the figures is much too small. They are not readable in the print out.

We thank the referee for pointing this out and we have enlarged the labels of the figures in the revised manuscript.

I also think the title is much too general - though admittedly fancy.

This echoes a similar comment by Reviewer 1, and as stated above, we have changed the title to "Ultrafast Isomerization in Acetylene Dication After Carbon K-shell Ionization: To Be or Not to Be" in the revised manuscript.

Experiments by i.e. the Kling group (ref. 10) and by the Kitzler group reported isomerization of acetylene in the dicationic state. In their experiments the relevant dynamics occurred in the lowest triplet Π state, which has a much lower barrier and a higher lying transition state structure at the geometry of the neutral acetylene. Thus its barrier is easier to be overcome. It would also be interesting to discuss the possibility of spin orbit coupling from the relevant singlet states to the triplet states, especially the triplet Π state. This might open a more effective route to isomerization.

We agree with the referee that intersystem crossing (ISC) between the singlet and triplet states is possible. However, this would take considerably longer than 100fs because the spin-orbit coupling will be negligible for a molecule containing only C and H atoms. ISC on the 100fs time scale is possible in molecules with heavier O or S atoms (recall that SOC scales with the fourth power of the nuclear charge). However, even when O and S atoms are involved, the probability of ISC on short time scales is rather low (less than 10%). Thus, it is extremely unlikely that appreciable ISC occurs on the relevant time scale in the experiment we are modeling. We did calculate SOC matrix elements for acetylene dication using CASSCF, and we find the largest matrix element connecting the $1\Delta_g$ and 3Π states to be 3.21 meV, which corresponds to an estimated time scale for ISC of more than 1ps. For further calibration, we point out that a largest SOC matrix element almost 10x larger (21.3 meV) is required for ISC of ~3% of the population within 100fs in a recent study of an O-containing nucleobase (Richter, Mai, Marquardt and Gonzalez, *Phys. Chem. Chem. Phys.* **16** 24423 2014).

We have added some discussion about ISC in acetylene dication dynamics in Sec. IIIC of the supplemental materials of the revised manuscript.

Reviewer 3:

The manuscript analyzes possible mechanisms of isomerization of acetylene dication to vinylidene dication by ultrafast H migration, which was concluded to occur within 100 fs based on recent Coulomb explosion experiments. This experimental observation is at odds with theoretical calculations of the potential energy surfaces and rate constants for isomerization of acetylene dication in low lying electronic states, which showed that the H

migration should be much slower. The authors of the present work succeeded in resolving this mystery. They computed the dynamics of acetylene dication produced by Auger decay following X-ray photoionization of the carbon atom K shell; these nuclear dynamics calculations involved several electronic states of the dication. The results show that the experimental observation can be assigned in terms of decomposition of the dication into two fragments ($\text{CH}^+ + \text{CH}^+$ or $\text{C}_2\text{H}^+ + \text{H}^+$) involving rotation of the departing fragments relative one another, rather than in terms of the H migration. The fast H migration channel involving a non-adiabatic transition between different electronic states has been also found but the authors show that probability of this process is rather low and it cannot be responsible for the experimental observation.

The calculations have been carried out at a state-of-the-art level and the results are convincing. There are two important contributions of this work to the field: 1) the seeming contradiction between theory and experiment on the rate of H migration in acetylene dication, which in fact was present in the literature for the last two decades, has been resolved and 2) the authors demonstrated a theoretical approach for better interpretation of experimental data derived from the widely used Coulomb momentum imaging method.

On these grounds, I believe that this manuscript merits publication in Nature Communications.

I have one minor comment for the authors:

References 5 and 23 are identical.

We thank the reviewer for his positive comments. We also thank him for pointing out this typographical error, which we have corrected.

We believe we have addressed all the reviewer comments and hope that the revised text is now suitable for publication in Nature Communications. Please do not hesitate to contact me if there are any questions.

REVIEWERS' COMMENTS:

Reviewer #1 (Remarks to the Author):

The authors properly addressed all of the reviewer's comments in the revised manuscript. The paper reports new theoretical studies of ultrafast isomerization in acetylene dication, which will motivate further experimental and theoretical efforts for a deeper understanding of molecular dynamics in the X-ray wavelength range. I recommend the publication in Nature Communications.

Reviewer #2 (Remarks to the Author):

I am happy with the changes made in answer to my questions and recommend the paper for publication.

Only one final remark on SOC's. Although the calculated SOC between triplet and singlet maybe small, the effective coupling can become large, when the two states involved cross or become degenerate for several geometries along the reaction path or in general if the density of states is high. In this case the ΔE value in the denominator of the effective SOC coupling approaches zero and its value becomes large. That may not happen in the case of acetylene dication.

Reviewer 2:

I am happy with the changes made in answer to my questions and recommend the paper for publication.

Only one final remark on SOC's. Although the calculated SOC between triplet and singlet maybe small, the effective coupling can become large, when the two states involved cross or become degenerate for several geometries along the reaction path or in general if the density of states is high. In this case the ΔE value in the denominator of the effective SOC coupling approaches zero and its value becomes large. That may not happen in the case of acetylene dication.

We have softened our statement in the supplementary materials (Supplementary Note 3C):

Supplementary Page 15: This ~~rules out non-negligible effect of~~ makes intersystem crossing from the singlet to triplet states in acetylene dication dynamics quite improbable on the sub-100fs time scale.